# A Review of Nitrogen Removal for Urban Stormwater Runoff in Bioretention System

**Manal Osman** [1,2,*] , **Khamaruzaman Wan Yusof** [1], **Husna Takaijudin** [1], **Hui Weng Goh** [3],
**Marlinda Abdul Malek** [4], **Nor Ariza Azizan** [3] , **Aminuddin Ab. Ghani** [3]
**and Abdurrasheed Sa'id Abdurrasheed** [1]

[1] Department of Civil and Environmental Engineering, Universiti Teknologi PETRONAS,
Bandar Seri Iskandar 32610, Perak, Malaysia; khamaruzaman.yusof@utp.edu.my (K.W.Y.);
husna_takaijudin@utp.edu.my (H.T.); abdurrashee_16000331@utp.edu.my (A.S.A.)

[2] Department of Agricultural Engineering, University of Bahri, Khartoum-North 12217, Khartoum, Sudan

[3] River Engineering and Urban Drainage Research Centre (REDAC), Engineering Campus, Universiti Sains
Malaysia, Seri Ampangan, Nibong Tebal 14300, Penang, Malaysia; redac_gohhuiweng@usm.my (H.W.G.);
ariza_azizan@usm.my (N.A.A.); redac02@usm.my (A.A.G.)

[4] Institute of Sustainable Energy (ISE), Universiti Tenaga National, Kajang 43000, Selangor, Malaysia;
marlinda@uniten.edu.my

\* Correspondence: manal_17005662@utp.edu.my; Tel.: +60-1671-955-48

**Abstract:** One of the best management practices (BMPs) for stormwater quality and quantity control is a bioretention system. The removal efficiency of different pollutants under this system is generally satisfactory, except for nitrogen which is deficient in certain bioretention systems. Nitrogen has a complex biogeochemical cycle, and thus the removal processes of nitrogen are typically slower than other pollutants. This study summarizes recent studies that have focused on nitrogen removal for urban stormwater runoff and discusses the latest advances in bioretention systems. The performance, influencing factors, and design enhancements are comprehensively reviewed in this paper. The review of current literature reveals that a bioretention system shows great promise due to its ability to remove nitrogen from stormwater runoff. Combining nitrification and denitrification zones with the addition of a carbon source and selecting different plant species promote nitrogen removal. Nevertheless, more studies on nitrogen transformations in a bioretention system and the relationships between different design factors need to be undertaken.

**Keywords:** stormwater runoff; bioretention; nitrogen removal; leaching

## 1. Introduction

Urban areas are constantly expanding in terms of space and density [1], with their population around the world expected to rise by 66% in 2050 [2]. As a result of population growth and urbanization, water pollution has increased exponentially [3]. Furthermore, stormwater runoff has a considerable impact on water pollution. It has long been recognized as a source of nonpoint source pollutants [4,5]. Excessive nitrogen pollution has been globally identified in a large portion of water bodies. Future land use activities are expected to intensify nitrogen loading [6]. Therefore, the prevalence of nitrogen has become a primary concern in stormwater management [7]. There are several alternatives for stormwater runoff control, namely filter strips, infiltration trenches, vegetated roofs, permeable pavement, rain gardens, bioretention, and swales [8]. Of these, a bioretention system is increasingly being used worldwide and is considered a good alternative for treating stormwater runoff [9]. The removal efficiency of stormwater pollutants in this system is generally satisfactory, except for nitrogen removal which is deficient in some operating systems [10–13].

This paper summarizes the recent studies that have focused on nitrogen removal from urban stormwater runoff in bioretention systems. It also discusses the recent advances, performances, and influencing factors of a bioretenton system.

### 1.1. Stormwater Runoff Characteristics

In urban areas, water pollution is a major challenge leading to chemical, physical, and biological damage to the environment [4,14], and therefore it contributes to ecological degradation and health risks [14,15]. Water pollution can be classified into two sources, namely point source and nonpoint source. A point source pollution refers to any single specific source from which pollutants can be discharged such as wastes from sewages and industries [16]. It is regulated through the implementation of standards, systematic laws, and high-quality engineering measures [4,5]. On the other hand, nonpoint source pollution originates from different sources including agricultural runoff, atmospheric deposition, and urban stormwater [17]. It comprises inorganic pollutants including nitrogen, phosphorus, metals, and sediments, as well as organic pollutants including pesticides and pathogens. To date, point source pollution has been recognized as a leading cause of water pollution [18], representing approximately half of the total pollutants in the world, with 57% being nitrogen [19].

Stormwater can be defined as the runoff from pervious and impervious surfaces in urban areas [20]. It includes some sewer discharges, flow from impervious surfaces such roads and parking lots, and flow from open spaces and construction sites. Groundwater flooding also can act as a contributory source especially during heavy storm events [21]. As runoff accelerates from these lands, it carries more pollutants to water bodies and increases loading of toxic contaminants. Excess pollutants impact water quality when water and soil containing pollutants wash into nearby waters or leach into ground waters [22,23]. There are two types of stormwater pollutants, namely gross pollutants and dissolved pollutants. Gross pollutants include sediments of different sizes such as vegetation, plant debris, paper, plastic, and others, and dissolved pollutants include nutrients, heavy metals, and hydrocarbons [10,24]. Pollution can also occur by direct runoff or by infiltration through the root zone which is then discharged to surface water [25]. There are typical pollutants characterizing stormwater runoff, with the most common pollutants being total suspended solids (TSS), nutrients including total nitrogen (TN), ammonium-nitrogen ($NH_4$-N), nitrate-nitrogen($NO_3$-N), nitrite-nitrogen ($NO_2$-N), total phosphorus (TP), and orthophosphate ($PO_4^{3-}$) [26]. The classification of pollutant load according to water quality standards [27,28] is shown in Table 1. In total there are five classes, namely Class I (clean water), Class II (moderately polluted), Class III (heavily polluted), Class IV (excessively polluted), and Class V (extremely polluted).

**Table 1.** Water quality standards.

| Parameter | Unit | Classes * | | | | |
|---|---|---|---|---|---|---|
| | | I | II | III | IV | V |
| **TSS** | mg/L | <25 | 25–50 | 50–150 | 150–300 | >300 |
| TP | mg/L | ≤0.05 | ≤0.15 | ≤0.6 | ≤1.2 | >1.2 |
| $PO_4^{3-}$ | mg/L | ≤0.02 | ≤0.1 | ≤0.4 | ≤0.8 | >0.8 |
| TN | mg/L | ≤1 | ≤3 | ≤12 | ≤24 | >24 |
| $NH_4$-N | mg/L | ≤0.04 | ≤0.3 | ≤1.2 | ≤2.4 | >2.4 |
| $NO_3$-N | mg/L | ≤1 | ≤2.5 | ≤10 | ≤20 | >20 |
| $NO_2$-N | mg/L | ≤0.01 | ≤0.1 | ≤0.4 | ≤0.8 | >0.8 |

* Class 1, excellent; class II, good, conventional treatment is required; class III, extensive treatment is required; class IV, for major agricultural activities which may not cover minor application to sensitive crops; class V, Bad which do not meet any of the above-mentioned classes.

Nitrogen represents the highest rated nutrient in stormwater runoff [7] and its concentration depends on the land use activities such as residential, parking lots, highways, commercial areas, and agricultural lands [6]. Figure 1 shows the variation of nitrogen concentration associated with different land use activities and it clearly shows that the higher concentration of nitrogen is found from emissions and agricultural activities. Emissions are produced from fluid leaks from vehicles, i.e., high density traffic areas, highways, and urban areas. The agricultural activities ultimately result in a greater load of sediment and nutrient accumulating in water bodies. In agricultural lands, pollution is primarily caused from fertilizers, herbicides, pesticides, and insecticides, all of which are rich in nitrogen [29].

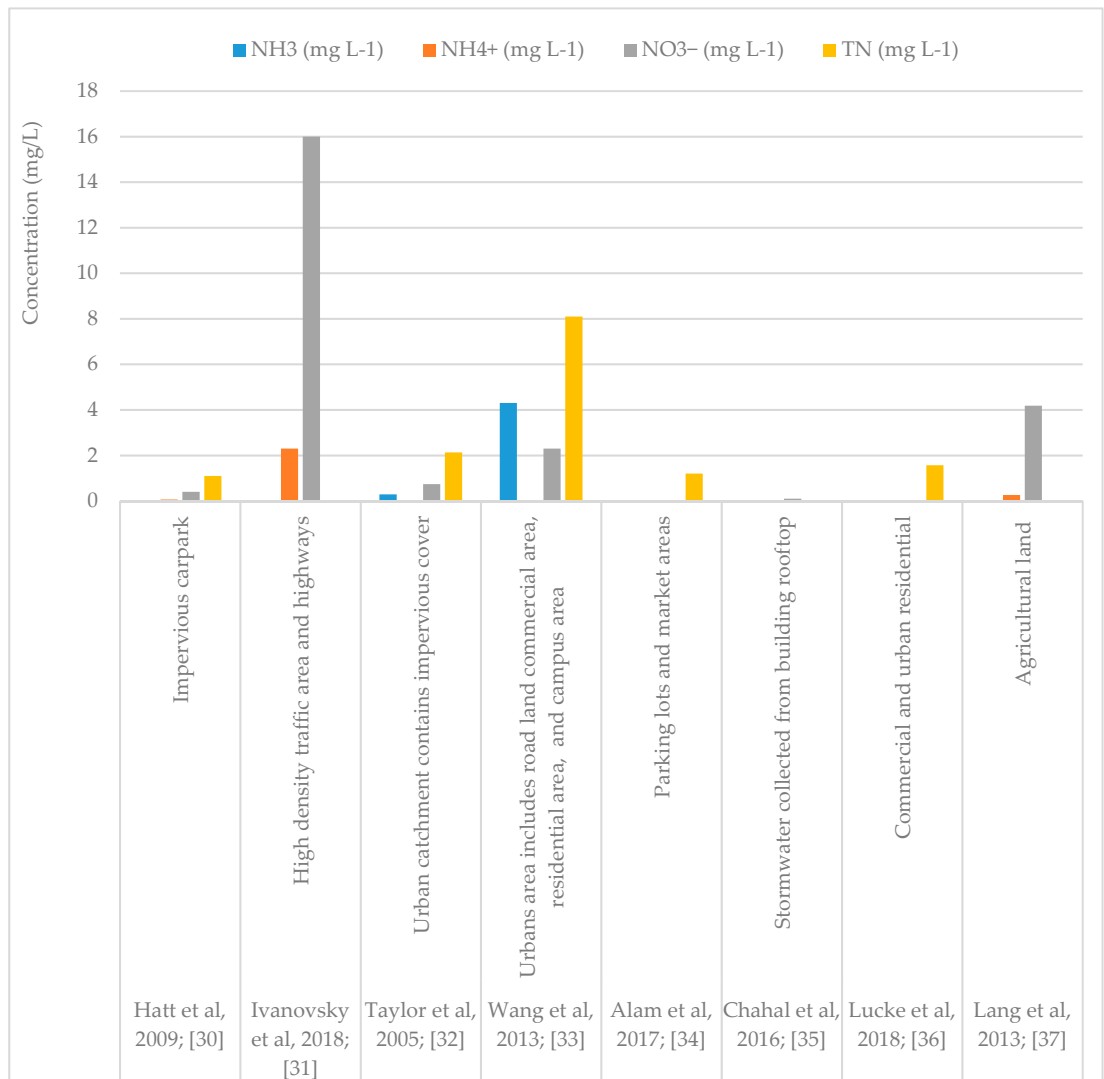

**Figure 1.** Summary of available data on nitrogen concentrations in stormwater runoff from several studies [30–37].

### 1.2. Best Management Practices (BMPs) and Low Impact Development (LID)

In order to minimize the environmental impacts of water pollution, it is necessary to establish water quality monitoring and intelligent watershed management [1,6,20,22,31,38,39]. Low impact development (LID) and best management practices (BMPs) are two innovative water management approaches to manage runoff as close as possible to its source [40]. Both have recently attracted interest from urban planners and researchers. To date, there have been various practices with regards to LID and BMPs for stormwater runoff control. Such practices include filter strips, infiltration

trenches, vegetated roofs, permeable pavement, rain gardens, bioretention, and swales [8]. Stormwater management could be divided into two main functions, stormwater quantity control and stormwater quality control. Stormwater quantity control is measured to curb flooding problems by facilitating detention to reduce the peak flow rate through the temporary storage and slow release of stormwater over an extended detention time. The main objective of stormwater quantity control is to prevent the occurrence of flash floods. Stormwater quality control, however, is intended to reduce nonpoint source pollution problems, whereas the primary objective of stormwater quality control is to achieve good water quality standards [41]. The use of these practices for water quality control is believed to be cost effective [42,43]. They take advantage of natural processes such as infiltration to reduce the volume and rate of runoff, while at the same time improving water quality [44,45]. The advantages of the infiltration process include groundwater recharge, runoff volume reduction, low stream flow augmentation, and water quality enhancement [46]. In these practices, flood mitigation can be achieved by promoting stormwater infiltration, storage, percolation, and evapotranspiration. Soil promotes water infiltration, storage, and percolation, whereas soil and vegetation together help to remove water through the evapotranspiration process. In the evapotranspiration process, water is transferred to the atmosphere through evaporation from the soil surface and transpiration by the plant. Pollutant mitigation can be achieved by allowing stormwater to be treated by vegetation and soil, thereby possessing complex chemical reactions as well as physical and biological processes [41,47].

LID and BMPs reduction targets for stormwater pollutants have been classified into categories in order to evaluate the performance of these practices in terms of stormwater quality control. For example, the pollutant reduction targets according to LID and BMPs in Malaysia were classified into three main categories, i.e., low, medium, and high [26]. The classification of reduction targets for the most common pollutants in stormwater runoff in Malaysia is shown in Table 2.

**Table 2.** Classification of reduction targets according to LID and BMPs in Malaysia.

| Pollutant | Low | Medium | High |
|---|---|---|---|
| TSS | Less than 40% of particulates greater than 0.125 mm retained | 40%–70% of particulates greater than 0.125 mm retained | >70% of particulates greater than 0.125 mm retained |
| Nutrients (TN & TP) | Less than 10% reduction | 10%–40% reduction | >40% reduction |

*1.3. Bioretention as a Promising BMPs and LID*

A bioretention system is part of stormwater best management practices (BMPs) for stormwater quantity and quality control. In recent years, interests in bioretention systems for stormwater quality treatment have piqued [5,20,24,31,36]. They are typically used to treat stormwater that has run over pervious and impervious surfaces in urban areas. A bioretention system is easily defined as the process in which biological processes and rapid infiltration occur along with the storage of water to reduce pollutants [48–50]. It can be a good process to treat runoff as it maximizes water storage, and therefore water can be infiltrated easily [51]. It also reduces runoff volume which reduces pollutants [52,53]. The facility size for a bioretention system is often designed for treating the first flush of stormwater [51]. Water quality enhancements can be achieved through infiltration and sedimentation. Filtration through vegetation is the primary mechanism for pollutant removal followed by the settling of particles and infiltration into the subsurface zone. As runoff travels through the system, the vegetation reduces peak velocity while infiltration reduces flow volume, which promotes pollutant removal [54]. In addition to direct plant nutrient uptake, the vegetation increases microbial activity through nitrifying and denitrifying processes which lead to increased nutrients removal [55]. A bioretention system comprises basins or trenches that are filled with porous media and planted with vegetation for treating stormwater runoff. Bioretention media consists of different layers which mainly include a filter media layer, a transition layer, and a drainage layer, as shown in Figure 2. The filter media layer is generally

composed of sand mixed with small amounts of silt, clay, and organic matter (mulch) [33]. The organic matter has several functions including retaining moisture in the plant root zone, providing a medium for biological growth and decomposition of organic matter, and offering some filtration of pollutants, as well as protecting the soil bed from erosion [56]. The transition layer of sand is recommended to prevent the filter layer from being washed into the drainage, and to provide an additional detention medium. The drainage layer can be either gravel or coarse sand with sufficient hydraulic conductivity to allow infiltrated water to flow towards the underdrain and also to facilitate saturation conditions. A bioretention system operates by filtering the stormwater through a vegetated surface and then percolating the runoff through different filter layers where the extended treatments can take place. During percolation, pollutants can be mitigated by different processes including adsorption, infiltration, and some chemical and biological processes [38].

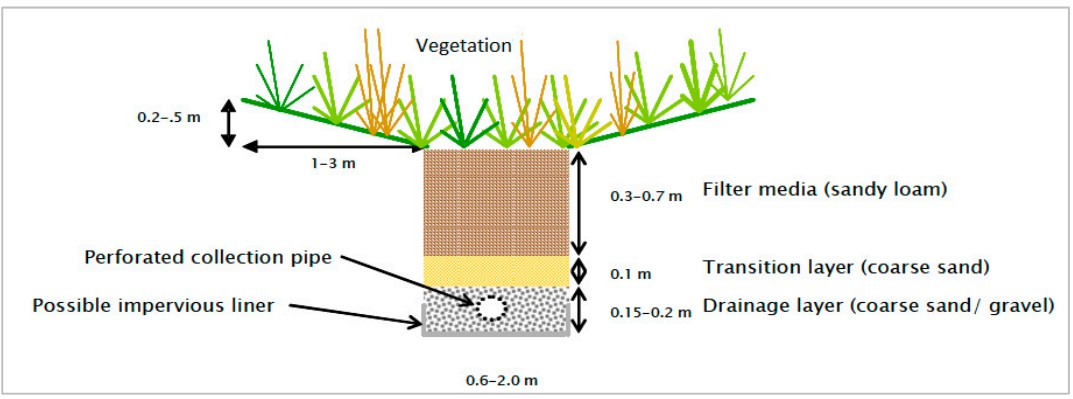

**Figure 2.** Bioretention [38].

## 2. Nitrogen Removal Processes

Nitrogen compounds include both organic and inorganic forms that are indispensable for biological life. The most important inorganic forms of nitrogen are ammonium ($NH_4^+$), nitrate ($NO_3^-$), and nitrite ($NO_2^-$). Gaseous nitrogen may exist as dinitrogen ($N_2$), nitrous oxide ($N_2O$), nitric oxide ($NO_2$ and $N_2O_4$), and ammonia ($NH_3$). The organic forms are dissolved organic N, and particulate organic N [57,58]. Nitrogen is primarily present in stormwater as an organic and a dissolved form [59]. Nitrogen has a complex biogeochemical cycle and is more difficult to remove because it is highly soluble [30,60]. Thus, the removal processes of nitrogen are typically slower than the removal of other pollutants [60]. An efficient removal of nitrogen is significantly dependent on physical processes, biological processes, and chemical reactions [61]. The main processes include assimilation (as N uptake), adsorption, mineralization (ammonification), nitrification, and denitrification [62,63]. Nitrogen removal processes always occur at varying rates [64]. Assimilation is the process in which inorganic nitrogen forms ($NH_4^+$, $NO_2^-$, and $NO_3^-$) are transformed into plant biomass by microorganisms and stored as organic nitrogen [6]. This organic nitrogen is temporarily stored in plant tissues and may be released again by decaying plants [63,65]. Mineralization is the conversion of organic nitrogen to ammonium ($NH_4^+$) [5]. Nitrification, on the other hand, is usually defined as the biological oxidation of ammonium to nitrate with nitrite [66].

$$2NH_4^+ + 3O_2 \rightarrow 2NO_2^- + 2H_2O + 4H^+ \tag{1}$$

$$2NO_2^- + O_2 \rightarrow 2NO_3^- \tag{2}$$

Denitrification is the process where nitrate is converted into dinitrogen gas ($N_2$) which is later released into the atmosphere or fixed to the plant roots [65,67]. A process called anaerobic ammonium

oxidation (anammox) was recently discovered in the 1990s [68]. It involves ammonium oxidation to $N_2$ gas using $NO_2$ in anoxic conditions. The simplified nitrogen cycle is shown in Figure 3.

$$NO_3^- \Rightarrow NO_2^- \Rightarrow NO \Rightarrow N_2O \Rightarrow N_2 \tag{3}$$

$$6CH_2O + 4NO_3^- \rightarrow 6CO_2 + 2N_2 + 6H_2O \tag{4}$$

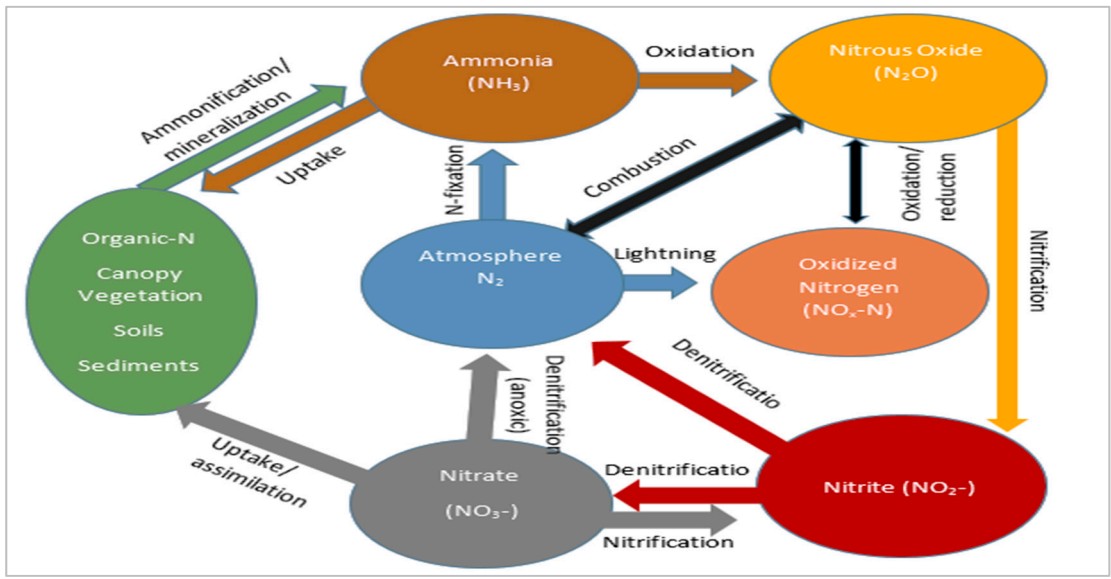

**Figure 3.** Simplified nitrogen cycle.

Nitrogen removal is a major area of interest for stormwater quality control. Previous works have intensively focused on nitrogen removal in a bioretention system [9,66,69]. Nitrogen removal in a bioretention system is always variable and mainly dependent on factors such as vegetation, soil filter media, influent concentration, and hydraulics factors [10,62,67].

## 2.1. The Effect of Vegetation

Vegetation is an essential component of a bioretention system which facilities oxygen transport to the soil and enhances the biological processes. Vegetation plays a crucial role in the performance of nitrogen removal [62,70]. Previous studies have shown that a vegetated bioretention system removes a greater amount of nitrogen than a nonvegetated bioretention system [10,67,70]. As runoff travels through the system, the vegetation reduces peak velocity while infiltration reduces flow volume, and thus pollutant removal is promoted [54]. Barrett et al. [10] compared the pollutant removal efficiency of bioretention systems for different media and plant species. The results showed a significant improvement in nutrient removal by a filter with the presence of plants. The presence of vegetation enhanced nitrogen removal and had a significant effect on TN and $NO_x$ removal [10,70]. A more recent study on nitrogen removal from stormwater runoff in mountainous cities was conducted by Wang et al. [67] using different stepped bioretention systems with different plant species. The results showed successful nitrogen removal. It was confirmed that the plant species play an important role for nitrogen removal. There are wide variations in nitrogen removal by different plant species in bioretention systems [9,11,30,50,66,69] as shown in Appendix A, Table A1. The variation among plant species in nitrogen removal is due to differences among species in plant size and plant uptake [65,71]. Each plant species, in different growth stages, has a different uptake rate. The plant uptake contributes towards $NH_4^+$ and $NO_3^-$ removal and is an important mechanism for $NO_3^-$ removal. $NO_3^-$ retention occurs through two mechanisms, plant uptake and denitrification. The uptake rate usually depends on the plant type and plant growth rate. In fact, good plant growth with higher biomass will result in a

better plant uptake and nitrogen removal [72–76], however, rapid plant growth rate also contributes to TN removal [77]. A study by Milandri et al. [78] found that the rapid growth rate of the turf grasses, Pennisetum and Stenotaphrum, was effective in $NH_3$ (97%) and $NO_3^-$ removal (>80%). A study by Chen et al. [79] showed that TKN concentration was significantly higher in the leaves and roots as compared to the stems of the *Elateriospermum tapos* plants. Plant roots also affect nitrogen removal, since thicker roots can take up a large amount of nitrogen [77]. In addition to direct plant nutrient uptake, vegetation can enhance microbial activity, and thus nitrogen removal [55].

## 2.2. The Effect of Soil Filter Media

Soil filter media play an important role in the removal of pollutants from stormwater [80]. Soil filter media supply plants with a suitable medium for growth and provide a well-drained soil [81]. Several studies have focused on nitrogen removal using different bioretention media. Lintern et al. [82] emphasized that TN removal efficiency was more affected by soil media depth for shallow rooted plant systems. Two studies conducted by Davis et al. [11,12] focused on the removal efficiency of nutrients from synthetic stormwater runoff using shallow bioretention depths. Their results indicated good removal efficiency of total nitrogen while nitrate reduction was poor. Soil media depth played a significant role in N removal. A deeper soil layer with low infiltration rate was needed to provide more detention time. It could enhance nitrogen removal by reducing the peak flow through storing water within the filter media layers. Increasing soil depth offered better removal efficiency of nitrogen [83]. Deeper soil media were more effective for ammonia removal due to increasing contact time, and thus enhanced the nitrification process [84,85]. There was, however, no significant effect of filter media depth on nitrate removal, which could be attributed to the limited denitrification process by contact time under anoxic conditions [86]. It is recommended that a bioretention system should contain 50% to 60% sand and 40% to 50% mixture of loam or sandy loam or loamy sand. Clay content should be 5% to 8% because too much clay can reduce infiltration into the soil [73].

## 2.3. The Effect of Nitrogen Concentration

Nitrogen influent concentration is one of the factors that affects nitrogen removal processes. Previous studies have shown that the uptake rate of nitrogen was influenced by the inflow concentration; the removal efficiency was more satisfied with the low nitrogen concentration than the high concentration [72,73]. S. Wang [67] observed that the removal efficiency of $NO_3^-$ was low (20.5%) for an influent concentration ranging from 6.15 to 9.61 mg/L. Moreover, TN removal efficiency was poor (~15%) for an influent concentration ranging from 10.23 to 14.11 mg/L. The low concentration of $NO_3^-$ could be effectively removed through the denitrification process, whereas $NH_4^+$ at different concentrations in the influent could be significantly removed by the bioretention system which may be attributed to the removal by the adsorption process through the bioretention media [66,87]. An investigation carried out by Bučiene and Gaigalis [88] revealed that nitrogen concentration in effluent was positively correlated with its concentration in the runoff. In addition, $NO_3^-$ concentration in the effluent was linearly increased with an increasing TN concentration in the effluent [89]. Furthermore, a comprehensive load reduction was always better under a lower concentration than under a high concentration [84].

## 2.4. The Effect of Hydraulic Factors

The most important hydraulic factors affecting nitrogen removal in a bioretention system are flow rate, runoff volume, retention time, and hydraulic conductivity [90,91]. As early as 1978, a study that characterized urban runoff by Whipple et al. [16] revealed that nutrient loadings in the effluent are directly proportionate to the flow rate of the runoff. The ability of a bioretention system to treat high stormwater runoff volume is relatively low [92]. Therefore, better nitrogen removal can be achieved with a low runoff volume rather than a high runoff volume [70], due to low stormwater runoff volume being effectively captured by the bioretention system. Meanwhile, high stormwater runoff volume may bypass the system without achieving the desired treatment target [92]. Retention times in the

bioretention system need to be adjusted accordingly to prevent nitrate leaching to the effluent [93]. A bioretention system with low infiltration rates can effectively remove $NH_4^+$ with sufficient retention time [94]. Increasing retention time can significantly improve nitrate removal and, consequently, enhance nutrient removal efficiency [90]. The denitrification process needs longer retention time to allow more nitrate removal [56,95]. The recommended hydraulic conductivity in bioretention media is within 13 to 200 mm/hr. Higher hydraulic conductivity will not maintain soil moisture for sustainable plant growth, whereas lower hydraulic conductivity will not allow for runoff capturing [96].

## 3. Nitrogen Leaching

Numerous studies have indicated that a bioretention system tends to be an effective stormwater treatment [10–12,69], however, nitrogen leaching has been observed by many researchers [82,97–99]. Nitrogen is mainly leached in dissolved forms such as $NH_4^+$ and $NO_3^-$ [59,100]. Organic N can also be leached due to decomposition of the dead plants [63]. This occurs when assimilation exceeds the mineralization process [68]. Ammonia is continuously released due to the mineralization of organic nitrogen to ammonia [5,11]. Bioretention exports $NH_4^+$ and $NO_3^-$ because of the large nutrient content of organic matter used within soil media [35,101]. The accumulation of organic matter may also contribute to leaching in organic N [101,102].

In addition, some studies have shown that higher nitrogen load discharged from a bioretention system is due to nitrate and nitrite [13,99], as it is difficult to separate soluble nitrate and nitrite from water through the filtration process. The studies suggest that the change in chemical species from one to another occurs simply during infiltration [50]. Nitrate leaching is always higher than ammonia due to the negative charge of the nitrate ion, whereas ammonia has a positive charge which interacts easily with the media. Therefore, $NO_3^-$ concentration in the effluent is always higher than $NH_4^+$. Some nitrate leaching is assigned to the accumulation of the nitrate that resulted from the nitrification process [87,92]. which is also an indicator of a low denitrification process [65]. Nitrate leaching is increasing over time [35,101], however, temperature has also shown a clear effect on nitrate leaching. In cold areas (2 to 20 °C), the nitrogen removal was poor and leaching was observed and this increased as the temperature decreased [103]. The percentage of nitrogen leaching in different bioretention systems is shown in Table 3.

**Table 3.** Percentage of nitrogen leaching in different bioretention systems.

| System Description | $NH_4^+$ (%) | $NO_3^-$ (%) | TN (%) | Reference |
|---|---|---|---|---|
| Bioretention planted with different types of water tolerant plants | −39 | −384 to −57 | −48 | [13] |
| Bioretention filled with sandy loam soil and shredded wood and planted with different plant spices | | (−205) ± 181 | | [5] |
| Bioretention planted with high diversity and low-diversity plant- mix of iron and aluminum oxide | | −46 | −14 | [104] |
| Bioretention amended by compost | | −37 to −216,000 | | [35] |
| Bioretention with no saturation zone planted with Microlaena stipoides and Dianella revoluta | | −300 to −400 | | [105] |
| Bioretention box filled with a sandy loam soil and topped with a thin layer of mulch with different plant spices | | (−73) ± 18 | | [11] |

## 4. Design Features that Enhanced Nitrogen Removal

Nitrogen removal is mainly dependent on the nitrification and denitrification processes [104]. In the nitrification process, $NH_4^+$ removal occurs in an aerobic condition. It is always obtained in the upper layer of bioretention media. In the conventional bioretention cell, the media layers are almost aerobic due to the high content of sand. $NH_4^+$ can also be removed by adsorption through soil layers [50,98] and nitrogen removal by denitrification accounts for 79.5% of total nitrogen removed [106]. A portion of nitrate removal can be achieved by the sorption process through bioretention soil media [107].

This removal is often insufficient due to a lack of denitrification [105]. An aerobic condition is likely to increase nitrification, while at the same time, limit the denitrification process. Thus, this system is capable of removing ammonium and incapable of removing nitrate [108].

Poor nitrogen removal in bioretention systems has been reported by some researchers [97–99,105,109]. Therefore, nitrogen removal has become a major concern in recent studies. There have been many attempts to improve nitrogen removal efficiency, with different methods being used such as adding a carbon source and saturation condition [63,110,111]. Several studies have pointed to an advancement of nitrogen removal in a bioretention system amended by a carbon source [77,95,112–114]. Carbon source additives including newspaper, woodchips, compost, biochar, cockle shell, and coconut husk have been widely used. The use of a carbon source has been recommended for engineering designs to promote N transformation. It plays a significant role in N removal through maintaining plant growth, soil properties, absorption, infiltration, and retention [115]. Plant uptake is significantly improved by increasing plant growth. These additives have created small anoxic zones for a further nitrification process [63,77]. Biological denitrification is also improved [109] by creating anaerobic conditions in the soil media, which, subsequently, leads to efficient N removal [110]. High nitrate removal rates are observed with soil media containing higher organic matter [116]. In fact, the use of a carbon source has been proven to be successful in various systems [110,111,117,118], however, reducing the amount of carbon source additives in the soil media is also recommended to avoid N leaching [5,119], because excessive use of carbon source additives can sometimes be the source of nutrients leaching [35,101].

In some bioretention systems, poor nitrogen removal could be enhanced by retrofitting the saturated zone to create anaerobic conditions for an effective denitrification process. It is recognized that high removal efficiency of nitrogen requires the ability of the system to provide aerobic and anaerobic conditions to ensure good removal and avoid leaching [120]. An aerobic condition can be achieved through a soil filtration media layer, whereas an anaerobic condition requires a saturated zone to increase the bacterial activity for the denitrification process. A system with a saturation zone can effectively remove nitrate as opposed to a system without a saturation zone [121]. Increasing the saturation zone depth can significantly enhance ON and $NO_3^-$ removal. $NO_3^-$ removal rate is correlated with the saturation zone depth. By increasing the saturation zone depth from 0 to 600 mm, the $NO_3^-$ removal efficiency can be significantly increased, whereas $NH_4^+$ removal is not affected by saturation zone depth [122–124]. An anaerobic zone would have a remarkable effect on denitrification and present very little opportunity in the nitrification process, however, nitrification also can take place in an anaerobic zone [50]. The system combining a saturation zone and a carbon source performs better in nitrogen removal. It is very effective in increasing the denitrification process and improving plant growth [75]. Furthermore, the denitrification process is generally limited by contact time under anoxic conditions. As such, a deeper anoxic zone is needed for more denitrification process as it can provide greater detention time [56]. The denitrification rate will also increase with the supply of soil water content as it determines the oxygen transfer rate from the atmosphere to the site where biological degradation takes place [125,126]. As suggested by Klein and Logtestijn [125], the minimum volumetric water content for denitrification in loam soil should be 40%. Table 4 shows the different design features that have been used in previous studies to improve nitrogen removal in bioretention systems.

**Table 4.** Design features to improve nitrogen removal.

| Design Features to Improve Nitrogen Removal | TN (%) | $NH_4^+$ (%) | $NO_3^-$ (%) | Ranking | Reference |
|---|---|---|---|---|---|
| Bioretention column with less permeable soil layer | 82 | 83 | 84 | High | [94] |
| Wood chips | 88 | | | High | [127] |
| Saturation zone | 49.8 | | | Medium | [128] |
| Different depths of saturation zone | | 80 | 62 | Medium-high | [124] |
| Combination of saturated to unsaturated sequence | | | 91 | High | [129] |
| Newspapers | 80.4 | | | High | [130] |

**Table 4.** *Cont.*

| Design Features to Improve Nitrogen Removal | TN (%) | NH$_4^+$ (%) | NO$_3^-$ (%) | Ranking | Reference |
|---|---|---|---|---|---|
| Planted bioretention with saturation zone | 93 | 95 | 67 | High | [75] |
| Bioretention with biochar and poultry litter | | 90 | | High | [131] |
| Bioretention planted with vegetables | 47 | | | Low | [74] |
| Saturation zone containing shredded newspaper | | | 99 | High | [111] |
| A large-scale column study with different plant species, filter media types and depths, and pollutant concentrations | | 93 | | High | [70] |
| Box prototype bioretention system filled with sandy loam soil and mulch | | 60–80 | | Medium-high | [9] |
| Bioretention contains carbon source and anoxic zone | | 71.1 | | Medium | [63] |
| Two-layered bioretention system amended with wood chips | | | 80 | High | [110] |
| Bioretention columns with filter media contains 8% organic material | | | 60–90 | Medium-high | [119] |
| Saturated zone containing woodchips | 61.9 | | 82.4 | Medium-high | [95] |
| Bioretention amended with biochar | | | 30.6–95.7 | Low-high | [117] |
| Columns study for anoxic sand packed amended with wheat straw, wood chips, and sawdust | | | 95 | High | [132] |
| Saturated zone combined with carbon source | | | 85–94 | High | [133] |
| Bioretention amended with biochar coupled with saturated zone | 20–30 | 50–60 | 50–60 | Low-medium | [123] |
| Bioretention combined with saturated and unsaturated conditions | | | 42–63 | Medium- high | [90] |

The type of filter media has shown different effects on nitrogen removal [9,94]. For example, the removal efficiency of sandy loam soil for NH$_4^+$ ranged between 60% and 80%, whereas a less permeable soil performed better with a removal efficiency of approximately 83%, and using different plant species and increasing the depth of filter media substantially enhanced NH$_4^+$ removal up to 93% [70]. On the other hand, the addition of carbon source additives to the filter media has shown a beneficial influence by enhancing nitrogen removal efficiency [110,117,127,130]. Furthermore, the use of a saturated zone and increasing its depth has provided additional removal of nitrogen, particularly for NO$_3^-$ [75,124,133]. Significant improvement in nitrogen removal was achieved by combining the saturated and unsaturated zones [90,129]. In addition, amending the saturated zone with a carbon source significantly removed NO$_3^-$ [134], up to 99% [111].

Overall, advanced nitrogen removal in bioretention systems can be achieved by combining nitrification and denitrification conditions with the addition of a carbon source. It is considered to be an optimal method for enhancing nitrogen removal and a promising way for stormwater runoff treatment.

## 5. Conclusions

The current study has reviewed recent advances in nitrogen removal for stormwater runoff in bioretention systems. Various studies have indicated that a bioretention system tends to be effective in nitrogen removal, however, several studies have reported nitrogen leaching. Nitrogen has a complex biogeochemical cycle and is more difficult to remove as it is highly soluble. Thus, the removal processes of nitrogen are typically slower. It is mainly dependent on physical and biological processes and chemical reactions. The main processes include assimilation, adsorption, ammonification, nitrification, and denitrification. In conclusion, advanced nitrogen removal in a bioretention system can be achieved by selecting an appropriate design. Combining nitrification and denitrification conditions by adding a carbon source has shown a beneficial influence on promoting nitrogen removal. It is a promising way for stormwater runoff treatment as it effectively enhances nitrogen removal. Additionally, proper selection of plant species can facilitate nitrogen removal, particularly where nitrogen concentrations are of

critical concern. Nonetheless, more studies on nitrogen transformations through a bioretention system and factors affecting them need to be undertaken. The relationships between various design factors and their combined effects on nitrogen removal must be considered for better design optimization. In addition, greater focus is needed on the development of bioretention design criteria which can promise more nitrogen removal enhancements.

**Author Contributions:** Investigation, original draft preparation, writing, and formal analysis, M.O.; supervision and review, K.W.Y. and M.A.M.; supervision, review and funding, H.T. and M.A.M.; supervision, review and editing, H.W.G. and M.A.M.; funding and resources, M.A.M.; review and editing, N.A.A. and A.S.A.; project administration, A.A.G. and M.A.M.

**Funding:** This research was funded by the Universiti Tenaga National, Malaysia, iRMC Bold 2025, grant code (RJO10436494) and the Universiti Teknologi PETRONAS, Malaysia, YUTP grant (015LC0-151).

**Acknowledgments:** The authors would like to acknowledge the support given by the Universiti Teknologi PETRONAS, and the River Engineering and Urban Drainage Research Centre (REDAC), Universiti Sains Malaysia.

**Conflicts of Interest:** The authors declare no conflict of interest.

## Appendix A

**Table A1.** Summary of nitrogen removal (%) by different plant species in bioretention studies.

| 1. Field study | | | | | | | | | | | | | | |
|---|---|---|---|---|---|---|---|---|---|---|---|---|---|---|
| Type of plants used | $NH_3$ | $NH_4^+$ | $NO_2^-$ | $NO_3^-$ | TKN | TN | TDN | ON | DON | PON | Use of C source | Use of plant | Site Name | Reference |
| chokeberry (Aronia prunifolia), winterberry (Ilex verticillata), and compact inkberry (Ilex glabra compacta) | 82 | | | 67 | 26 | 51 | | 14 | | | no | yes | Haddam, Connecticut., US | Dietz and Clausen (2006) [135] |
| river birch (Betula nigra), common rush (Juncus effuses), yellow flag iris (Iris pseudacorus), sweetbay (Magnolia virginiana) | −1 | | | 75 | −5 | 40 | | | | | no | yes | Greensboro, N.C., US | Hunt et al. (2006) [50] |
| Southern wax myrtle (Myrica cerifera), Virginia sweetspire (Itea virginica), winterberry (Ilex verticillata) inkberry (Ilex glabra) | 86.0 | | | 13 | 45 | 40 | | | | | no | yes | Chapel Hill, N.C. US | Hunt et al. (2006) [50] |
| Blueflag iris (Iris virginica), cardinal flower (Lobelia cardinalis), common rush (Juncus effusus), hibiscus (Hibiscus spp.), red maple (Acer rubrum), sweet pepperbush (Clethra alnifolia), Virginia sweetspire (Itea virginica), wild oat grass (Chamanthium latifolium) | | 73 | | | 44 | 32 | | | | | no | yes | Charlotte, N.C., US | Hunt et al. (2008) [136] |
| red maple (Acer rubrum), sweet bay (Magnolia virginica), Virginia sweetspire (Itea virginica), liriope (Liriope sp.), verbena (Verbena sp.), and blackeyed Susan (Rudbekia hirti). | 74 to 82 | | | −209 to −477 | | −21 to −75 | | −2 to −8 | | | no | yes | Nashville, N. C., US | Brown and Hunt (2011) [86] |
| n/a | | | | | | 19.9 to 90.8 | | | | | no | yes | LTU, Southfield, MI, US | Carpenter et al. (2010) [137] |
| prairie cord grass (Spartina pectinata) sumpweed (Iva annua) | | | | 33 | | 56 | | | | | no | yes | Lenexa, Kansas, US | Chen et al. (2013) [66] |
| Creeping juniper plants | | | | 16 | 52 | 49 | | | | | no | yes | Greenbelt, Maryland | Davis et al (2006) [11] |
| Creeping juniper plants | | | | 15 | 67 | 59 | | | | | no | yes | Largo, Maryland | Davis et al (2006) [11] |
| Trees | | | | | | 58.6 | | | | | no | yes | KNU, Chungnamdo, Korea | Geromino et al. (2013) [138] |
| Dianella species, C. appressa | | 96.0 | | −17.0 | | 37.0 | | | 58 | 79 | no | yes | McDowall, Australia | Hatt et al. (2009) [30] |
| Carex appressa, Carex tereticaulis, Lomandra longifolia, Isolepis nodosa, Caleocephalus lacteus, and Juncus spp. | | 64 | | −13 | | -7 | | | −129 | 38 | no | yes | Monash University, Australia | Hatt et al. (2009) [30] |
| n/a | 82 | | 82 | −137 | | 9.7 | | | −146 | 83 | no | yes | College Park, Md., US | Li et al. (2014) [102] |
| n/a | | | | 86 | | | | | | | no | yes | College Park, Md., US | Davis (2007) [69] |

**Table A1.** *Cont.*

| Type of plants used | NH$_3$ | NH$^+$ | NO$_2^-$ | NO$_3^-$ | TKN | TN | TDN | ON | DON | PON | Use of C source | Use of plant | Location | Reference |
|---|---|---|---|---|---|---|---|---|---|---|---|---|---|---|
| grass | | 79.4 | | 43.1 | | 60.9 | | | | | no | yes | Piedmont of North Carolina | Smith and Hunt (2007) [139] |
| n/a | 77.4 to 78.7 | | | | | | | | | | no | no | Daxing District, Beijing, China | Liu et al (2017) [85] |
| Lomandra longifolia (Matt Rush) | | | | | | 11 to 75 | | | | | no | yes | Sunshine Coast, Australia | Nichols and Lucke (2016) [140] |
| hardy native perennials, shrubs, and trees | | | | | | 99 | | | | | no | yes | Blacksburg, Virginia | Debusk et al. (2011) [141] |
| | | | | 30.6 to 95.7 | | | | | | | yes | no | University of Delaware, Newark, DE, USA | Tian et al. (2019) [117] |
| n/a | 10 | | −56 | 9 | 37 | 25 | | | 53 | | no | yes | Balam Estate Rain Garden, Singapore | Wang et al. (2017) [142] |

**2. Laboratory study**

| Type of plants used | NH$_3$ | NH$^+$ | NO$_2^-$ | NO$_3^-$ | TKN | TN | TDN | ON | DON | PON | Use of C source | Use of plant | Type of study | Reference |
|---|---|---|---|---|---|---|---|---|---|---|---|---|---|---|
| Creeping juniper plants | | | | <20 | 55 to 65 | | | | | | no | yes | Pilot boxes | Davis et al. (2006) [11] |
| Carex rostrata Stokes (Bottle sedge) | | 51.7 | NOx = −1461 | | | −208 | | | −240 | | no | yes | Lab column | Blecken et al. (2007) [143] |
| Carex appressa, Melaleuca ericifolia, Microleana stipoides, Dianella revoluta, Leucophyta brownii | >93 | | NOx = 96 to −630 | | | 79 to −241 | | | | | no | yes | Lab column | Bratieres et al. (2008) [70] |
| Chrysanthemum zawadskii var. latilobum, Aquilegia flabellata var. pumila, Rhododendron indicum Linnaeus, Spiraea japonica | | 40 to 54 | | 35 to 41 | | 49 to 55 | | | | | no | yes | BR reactors | Geromino et al. (2014) [144] |
| | | | | | | 38 to −164 | | | | | no | no | Lab column | Hatt et al. (2008) [145] |
| Swamp Foxtail Grass (Pennisetum alopecurioides) Flax Lily (Dianella brevipedunculata), two woody shrubs, Banksia (Banksia integrefolia), Bottlebrush (Callistemon pachyphyllus) | | | NOx = 88 to 95 | | | 76 | | | | | no | yes | Lab column | Lucas and Greenway (2008) [76] |
| Twenty native plant species from Victoria and Western Australia and two common lawn grasses | | | | | | 58 to 89 | | | | | no | yes | Lab column | Payne et al. (2014) [146] |
| Monocots and Dicots | | −303.5 | NOx = 78.9 | | | −66 | −115.4 | | −509.9 | 21.6 | no | yes | Lab column | Read et al. (2008) [71] |
| Narrowleaf Blue-eyed grass (Sisyrinchium angustifolium) | | | | −1.14 | 60 | 36.4 | | | | | no | yes | Lab column | O'Neill and Davis (2012) [147] |
| | | | | | | >90 | | | | | yes | no | Pilot boxes | Kim et al. (2003) [109] |
| Buffalograss 609 and Big Muhly | | | NOx = −232 to 62 | | 65 to 89 | 59 to 79 | | | | | yes | yes | Lab column | Barret et al. (2013) [10] |

**Table A1.** *Cont.*

| | | | | | | | | |
|---|---|---|---|---|---|---|---|---|
| Carex appressa | 88 to 99 | NOx = 80 to 99 | 69 to 95 | | yes | yes | Lab column | Glastier et al. (2014) [148] |
| | | | 59.8 | | yes | no | Lab column | Guo et al. (2014) [149] |
| Twenty native plant species from Victoria and Western Australia and two common lawn grasses | | | 79 to 93 | | yes | yes | Lab column | Payne et al. (2014) [146] |
| Baumea juncea, Melaleuca lateritia, Baumea rubiginosa, Juncus subsecundus | 95 | NOx = 67 | 93 | 93 | yes | yes | Lab column | Zhang et al. (2011) [75] |
| Dianella revoluta, Microlaena stipoides and Carex appressa | 9.8 to 75.6 | NOx = −66.7 to 100 | −11.6 to 68.8 | −96.1 to 41.2 | yes | yes | Lab column | Zinger (2013) [105] |
| Creeping juniper plants | 60 to 80 | | 65–75 | | no | yes | Pilot boxes | Davis et al. (2001) [9] |
| Buxus Microphylla var. Koreana | | 97.5 | | | no | yes | Lab column | Cho et al. (2009) [87] |
| | 96.2 | | 86.4 | | no | no | Lab-scale (vertical tubes) | Yafei et al. (2017) [98] |
| Carex appressa. | | 90 | | | yes | yes | Lab column | Zinger (2007) [150] |
| Spinach (Ipomoea aquatic) | 64–78 | 68–89 | | | no | yes | Prototype system in green house | Endut et al. (2009) [151] |
| Turf-grass, succulent-perennial and reed | 90 | 69 | | | no | yes | Glasshouse | Milandri et al. (2012) [78] |
| Zoysia matrella | | | 42.6 | | no | yes | Lab column | Wu et al. (2017) [128] |
| Iris pseudacorus and Zoysia matrella | | | 49.8 | | no | yes | Lab column | Wu et al. (2017) [128] |
| n/a | >90 | 21 | 39 | | yes | yes | Lab column | Qiu et al. (2019) [152] |
| Ophiopogon japonicus and Radermachera hainanensis Merr. | >95 | 43.0–79.6 | 68.4–83.0 | | yes | yes | Lab column | Gongduan et al. (2019) [153] |

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
