# Peer review of "A Review of Nitrogen Removal for Urban Stormwater Runoff in Bioretention System"

_sustainability, doi:10.3390/su11195415_

Round 1
Reviewer 1 Report
This is a review of the manuscript entitled: ‘A review on nitrogen removal for urban stormwater runoff in bioretention system’ by Osman, Yusof, Takaijudin, Goh, Malek, Azizan and Ghani, whose intent is to summarized the recent studies focusing on Nitrogen (N) removal for urban stormwater runoff in bioretention system and discuss the recent advances in bioretention system. The factors affecting to enhance N removal (Vegetation, Soil filter media, N concentration and Hydraulic factors), and bioretention system design features that enhance N removal, were discussed.
As first consideration, the paper is unbalanced: there are 10 pages of text and 7 of references. I know that it is a review and therefore the number of references is important, but the text is badly organized: the authors write 4 pages of Introduction (Section 1) and only 5 more about factors that affect the nitrogen removal - which is the core of the paper - including conclusions.
In my opinion, the paper needs a lot of improvements to the current version in order to be considered for publication. More in detail please find below, section by section, my observations and comments.
Introduction
Line 33: non-point (and not nonpoint) as in the rest of the text;
Lines 35-36: even if the reference 7 is relevant, in my opinion it is misleading ... as stated by the authors "nitrogen pollution has been identified in the large portion of water bodies" therefore it cannot be defined as an exclusively concern in the saltwater management. Moreover, as the authors will certainly know, the major sources of nitrogen come from agricultural activities, and I expected to find some references about it, which I did not find;
Line 47: Correct the section title with 1.1 and not 2.1
Lines 66-68: The typical pollutants characterizing stormwater runoff (TSS, N, P, BOD and COD) are mentioned, but the sources of these pollutants are not described, which could be especially useful for a review and for readers, particularly in the choice of the most appropriate BMP or LID (beyond the bioretention system here analyzed;
Lines 75-76 and Figure 1: I suggest inserting an explanation of the sources of each nitrogen pollutant both for completeness of the revision and to better clarify the graph shown in the Figure 1;
Section 1.2 and 1.3: In general, these sections (1.1, 1.2 and 1.3) should be improved; many concepts are poorly written and exposed in a confused and approximate way ... often the same concepts are repeated in different parts of the text, citing studies without a common thread.
Line 80: Correct the section title with 1.2 and not 2.2;
Lines 82: I think it's necessary to insert some more References (over the 35 and 36);
Lines 83-85: I suggest to see and insert this Reference: Fletcher, T. D., Shuster, W., Hunt, W. F., Ashley, R., Butler, D., Arthur, S., ... & Mikkelsen, P. S. (2015). SUDS, LID, BMPs, WSUD and more–The evolution and application of terminology surrounding urban drainage. Urban Water Journal, 12(7), 525-542;
Lines 86-88: The authors state that "Water quantity control is measured to curb post construction flash flood problems while erosion and sediment control is measured to minimize erosion and sedimentation problems during construction"..authors should better conceptualize the distinction between "post- and during- construction" problems, that here it is not clear;
Line 95-96: The authors state that BMPs/LID solutions "also enhance pollutant mitigation by promoting water infiltration and evapotranspiration to improve water quality"...despite being known the benefits of infiltration processes for the improvement of quality, the authors should clarify those provided by ETP
Table 2: since the revision is not specific to the Malaysian territory, the authors should specify (not only by Reference number) that the standards shown in the Table 2 are not generalized but are only refer to Malaysia;
Line 99: I think is a journal editing problem, but I suggest moving the title of the section 2.3 to the other page. Moreover correct the section title with 1.3 and not 2.3
Lines 101-102: The sentence "In recent years, there has been an increasing interest in bioretention for stormwater quality treatment" needs to be supported with reference;
Line 109-110: The sentence “Then settling of particles, and infiltration into the subsurface zone” is incomplete;
Section 2: Section 2 (and related subparagraphs 2.1 to 2.4), as well as section 1, is approximate and not comprehensively described; it seems that the authors have included a number of sentences citing other studies, but without following a precise logical thread, except the title of the paragraph (and not even always).
Lines 147-149: Among the factors influencing nitrogen removal the authors cite vegetation, soil filter media, N concentrations and hydraulics factors, but I believe that there are other factors to be analyzed such as:
- Temperature dependence [look at Blecken G.T., et al. (2007). The influence of temperature on nutrient treatment efficiency in stormwater biofilter systems; Blecken G.T. et al. (2010). Laboratory study on stormwater biofiltration: Nutrient and sediment removal in cold temperatures]
- Structural Configuration [look at Jiang et al. (2017). Experimental study of nitrogen removal efficiency of layered bioretention under intermittent or continuous operation.]
- Saturated zone (wet and dry periods) [look at Wang et al (2018). Effect of saturated zone on nitrogen removal processes in stormwater bioretention systems ]
Line 162: The authors, referring to Reference 61, claim that "native species are more effective than exotic ones", but - as reported below - the reference cited affirms the contrary "no scientific results can support the hypothesis that native plants or diversely-planted systems offer better performance than systems planted with fewer species or with exotic species";
Table A1: Since the aim of this Table is to summarize and evaluate the different plant species for nitrogen removal in bioretention studies, I suggest to invert the column "Site Name" with "Type of Plants used" and introduce units ( I guess are concentrations in mg/l or %).
Furthermore since the table is very broad, I would suggest not to limit the distinction only between Laboratory Studies and On-Site Studies, but I suggest to make other sub-groups (eg. by geographical area, or even better, by type of plants used), so as to be able to better compare the analyzed studies and make the table more accessible to readers;
Lines 163-166: Observations and comments made in these lines should be (also) related to the table just shown (Table A1);
Lines 167-168: the sentence “The higher biomass and better plant growth; the higher plant uptake and better nitrogen removal”, as written, does not make sense;
Page 6 Section 2.3: Considering the topic of the revision, I find this section too short;
Lines 224-225: the sentence “The higher the nitrogen load of stormwater discharged from the bioretention was because of nitrite and nitrate”, as written, does not make sense;
Section 4: I thought it was a paragraph more focused on the different design/construction details that influence nitrogen removal (eg length/slope of the bioretention system, stratigraphy, layer thickness)...but, apart from Line 254, I find nothing about this.
Regarding different structural configuration (as layer characteristics, packing factors etc.) please look at Jiang et al. (2017). Experimental study of nitrogen removal efficiency of layered bioretention under intermittent or continuous operation.
Table 5: I find it useful this table but I do not find correspondence with what is written in paragraph ... I expected to find the same references or - at least - find those mentioned in the text. I would also suggest not limiting the analysis to a single study for each type of design feature but to compare different ones and/or create sub-groups;
Conclusions: As the same authors state (in Lines 300-301) “more work on nitrogen transformations through bioretention system and factors affecting them needs to be explored” and I think that this review was not enough and that much more work must still be done.
Author Response
Thank you for your great effort and valuable comments.
I have attached the correction
The red text means corrected or added.

Reviewer 2 Report
This paper is a comprehensive review of nitrogen (N) removal in stormwater bioretention systems. The authors have put together a nice collection of research on bioretention N removal with comparative tables on research methods and findings. However, the discussion of the work and the results needs to be improved.
For example, in section 2. Nitrogen removal processes-lines 134-137, the statements are unclear as to their meaning. Organic N once assimilated into plant biomass is only released when the plant biomass dies, becomes organic matter for mineralization by microbes. It is unclear how low denitrification can release organic N to the effluent. The conversion of organic nitrogen to ammonium nitrogen is mineralization, not magnification. A better discussion of these references may clarify the points made by the authors.
Lines 254 to 257: The discussion on additives to increase carbon content and N removal presents the findings of several studies but lacks the discussion of the mechanisms involved and how the systems were improved. A more thorough explanation of the research should be added to the section.
Lines 273-276: The authors need to explain what is meant by magnification. Do they mean an increase in concentration of nitrogen? If so, which form of nitrogen is increasing?
This paper pulls together a good number of studies on nitrogen removal in bioretention systems with nice comparative tables. However, discussion of the research results needs a more in-depth approach to make this work a significant contribution to the field.
Author Response
Thank you for your great effort to review this manuscript. Your corrections have really helped to improve it.

Reviewer 3 Report
With your manuscript you presented a very diligently written review of the stat-of-the-art of nitrogen removal from stormwater runoff by means of bioretention systems. Unfortunately, the draft suffers from a lot of insufficiencies. Many of them I have marked and discussed in the annotated text file attached as PDF. Please have a close look at this file. Additionally, I'd recommend reading the following book:
Handbook of Water Sensitive Planning and Design. Edited by Robert L. France, Lewis Publishers, CRC Press LLC, 2000 N.W. Corporate Blvd., Boca Raton, FL 33431, 2002. 699 p., ISBN 1-56670-562-2.
It will provide insight into the practical questions of stormwater management and nutrient retention, though possibly more directed at urban stormwater problems.
Generally missing in your draft are more recent European studies in rural environments, which have been induced by the European Water Framework Directive since 2000/2001.

Author Response
Dear reviewer thank you for your great effort and valuable comments.
The correction is attached. The red text means it corrected or added.

Round 2
Reviewer 1 Report
The authors thanked me for my “great effort and valuable comments” but they did not provide any response to individual comments except through the new version of the text (“red text means corrected or added”).
Apart from this observation, reading the new version of the text, I believe that - despite some minor changes in response to comments from other reviewers - the problems observed in the first version are not solved, and that my concerns remain unchanged.
The authors have changed some characters from lowercase to uppercase, have corrected the numbering of the paragraphs, following the observations of the other reviewers, and have partially explored some points, but the article has remained essentially the same. Some topics should have been deepened and thoroughly investigated (other factors influencing nitrogen removal; eg. Temperature, Structural Configuration, Saturated zone) and suggested references, that could have been useful to improve the article, were not taken into account.
In my opinion - as extensively explained in the first review - the paper is unbalanced (10 pages of text and 10 of references), poorly written and exposed in a confused and approximate way. Section 2, as well as Section 1, is approximate and not comprehensively described; it seems that the authors have included a number of sentences citing other studies, but without following a precise logical thread.
The paper, as it is has serious flaws and is not scientifically sound.
Author Response
We would like to thank the Reviewers for their prompt reply. We are particularly grateful for the constructive reviews of our work which have improved our manuscript. According to reviewer number one, we have tracked all the comments raised by him. Also, all the manuscript was proofread by an expert. regards.

Reviewer 2 Report
This paper is a comprehensive review of nitrogen (N) removal in stormwater bioretention systems. The authors have put together a nice collection of research on bioretention N removal with comparative tables on research methods and findings. This paper pulls together a good number of studies on nitrogen removal in bioretention systems with nice comparative tables.
It is difficult to see the tables as they will be printed, so be sure to set them up clearly.
The English grammar still needs some improvement with emphasis on verb tense and article use.
Author Response
I would like to thank the reviewer for the prompt reply. We are particularly grateful for the constructive reviews of our work that have improved our manuscript. According to reviewers' comments, we have tracked all the comments raised by him. Also, all the manuscript has been proofread by an expert. ​
Best regards.
Reviewer 3 Report
Great work done in cleaning your manuscript. It is accepted now without further remarks.
Author Response
Thank you for your effort and time to review this paper, your comments and input have really helped to improve the work. The proofreading and English editing by an expert ( as you stated) was considered to improve the paper and all manuscript was proofread.
My kind regards.
Round 3
Reviewer 1 Report
Revising the new version of the paper was not so easy because I received the draft copy (with track changes) and was difficult to find correspondence between the lines indicated by the authors and those actually corresponding to the text.
Despite this problem, reading the new version I found a substantial improvement compared to the original version thanks to the changes made based on the comments of the reviewers.
However in the Authors response to my revision, I have not found answers and justifications to all my comments (in my opinion often misunderstood and undervalued). In light of this, I think that the paper can be reconsidered for publication after minor changes to the current version.
All these changes are expaned in the text file attached (as PDF) and annotated in BLUE point by point.

Author Response
Thank you very much for your comments and suggestions, I very much appreciate. The comments and suggestions are valuable and very helpful for revising and improving the manuscript.
I'm especially grateful for the suggestions of reviewer 1 regarding the suggestion which have been very helpful in improving the manuscript.
Kind regards,
